# Global Path Planning of Unmanned Surface Vehicle Based on Improved A-Star Algorithm

**DOI:** 10.3390/s23146647

**Published:** 2023-07-24

**Authors:** Huixia Zhang, Yadong Tao, Wenliang Zhu

**Affiliations:** 1School of Ocean Engineering, Jiangsu Ocean University, Lianyungang 222005, China; dngfer970304@gmail.com; 2School of Mechanical Engineering, Jiangsu Ocean University, Lianyungang 222005, China; zhuwenliang@mail.chinamso.com

**Keywords:** unmanned surface vehicle, A-star algorithm, bi-directional search, path planning, path smoothing

## Abstract

To make unmanned surface vehicles that are better applied to the field of environmental monitoring in inland rivers, reservoirs, or coasts, we propose a global path-planning algorithm based on the improved A-star algorithm. The path search is carried out using the raster method for environment modeling and the 8-neighborhood search method: a bidirectional search strategy and an evaluation function improvement method are used to reduce the total number of traversing nodes; the planned path is smoothed to remove the inflection points and solve the path folding problem. The simulation results reveal that the improved A-star algorithm is more efficient in path planning, with fewer inflection points and traversing nodes, and the smoothed paths are more to meet the actual navigation demands of unmanned surface vehicles than the conventional A-star algorithm.

## 1. Introduction

An unmanned surface vehicle (USV) is an unmanned, autonomously navigated intelligent ship. Path planning plays a very important role as a part of the navigation system of the USV. It can optimize the motion trajectory of the USV, help the USV plan the optimal path, and improve the navigation accuracy, efficiency, and safety of the USV.

Path planning refers to the process of finding an optimal or specific path through an algorithm under a given map and a start-end location. The path planning of USV mainly involves two aspects: global path planning and local obstacle avoidance. At present, global path planning algorithms mainly include Dij algorithm [1], A-star algorithm [2], the PRM method [3], the RRT method [4], and intelligent bionomics path planning algorithms (examples include ant colony algorithm [5], and genetic algorithm [6]). Among them, the A-star algorithm is widely used in the fields of computational science and robotics research for its advantages of simplicity, speed, and adaptability in static global planning. In recent years, domestic and foreign researchers have achieved many results in optimizing and improving the A-star algorithm. Oluwaseun et al. [7] proposed an improved multi-objective A-star (IMOA-star) algorithm which is superior to the conventional A-star algorithm in processing time, path smoothing and path length; Hong Youkyung et al. [8] proposed an improved A-star algorithm combined with minimum capture trajectory generation to avoid collision with obstacles by extracting basic waypoints and certificated the algorithm performance through simulated and real experiments; Lima et al. [9] proposed an A-star algorithm based on dynamic simplification to calculate the optimal path of the mobile robot under real-time constraints. Gaofeng Yueet al. [10] proposed a path planning strategy based on the bidirectional smoothing A-star algorithm, which improves the real-time performance by optimizing the cost function and smoothing the planned path, greatly improving the correctness and feasibility of the algorithm.

As USVs are usually used in complex and harsh sea conditions and have characteristics such as fast speed and large inertia, their driving process is more likely to have reliability and safety problems. Chunyu Zhao et al. [11] added the collision function to the conventional A-star algorithm valuation function and optimized the neighborhood traversal method and node selection to increase the path smoothness and ensure the safety of unmanned ship navigation; Weinan Shu et al. [12] used the pixel thresholding method to eliminate redundant information and proposed a path optimization method for node obstruction detection method to ensure that the unmanned vessel is far away from obstacles and the algorithm traverses fewer nodes and path turning points; and Xin Chen et al. [13] enable the improved A-star algorithm to generate safe and reasonable global paths with less total mileage by combining the partition distance information and the angle variable weighting heuristic function.

Based on the optimization improvements and shortcomings of the current A-star algorithm applied to the aspect of USV, this paper makes the following improvements:Based on the conventional A-star algorithm evaluation function, we introduced a vertical distance node deviation factor based on the line connecting the initial point and the goal point to improve the heuristic function.We adopted the bidirectional search strategy to traverse the nodes from the initial point and the goal point at the same time.We smoothed the shortest path obtained from the planning combined with the B spline curve.

Consequently, we improve the efficiency of search of the algorithm and increase the path convergence smoothness.

## 2. Conventional A-Star Algorithm

### 2.1. Establishment of Environmental Mode

Setting up environmental model is the premise of path planning for USV. The common environmental modeling methods include grid method, geometry method, and topology method [14]. As a classical graph traversal path search algorithm, the A-star algorithm searches a certain area to plan a suitable path. The grid method can easily and effectively express the motion of the USV and simplify the calculation process of path planning.

The process of transforming the environment model into a grid map is mainly to divide the environment into small square areas, called grids, each grid represents a small part of the environment, and each grid is associated with the storage of the environment attributes (such as color, height, obstacles, etc.). As shown in Figure 1, the information of the unmanned ship environment is simplified into two color grids, where the white grid represents the unmanned ship navigable area and the black grid represents the non-navigable area (obstacle area), and the obstacle that does not occupy a full grid is inflated. Transforming the key information of the environment into rasterized data can facilitate computer analysis and operation, and improve the computer’s ability to sense the environment. We use two-dimensional raster maps to build the environment model of USV in this paper.

### 2.2. Selection of Search Neighborhood

Unmanned ship search neighborhood selection refers to the selection of goal points or areas within a certain area around the unmanned ship for autonomous navigation, path planning, and mission execution in unmanned ship control. Considering the complexity of various continuous movements of different unmanned ships in the real environment, the search neighborhood of the unmanned ship is discretized into 4, 8, 16, and other directions, called 4-neighborhood search, 8-neighborhood search, and 16-neighborhood search, etc., as shown in Figure 2.

Among them, in the 4-neighborhood search, we take the current grid as the center and search in four directions: up, down, left and right. Since only four directions are considered, its computation is small and easy to implement with fast computation speed, but the search accuracy is low and there may be inaccurate recognition at complex edges or corners. The 8-neighborhood search, on the basis of 4-neighborhood search, adds four diagonal directions including upper left, upper right, lower left, and lower right, which is more effective in dealing with some local details than the 4-neighborhood search, and is enough to find the location of the goal point more accurately. As for the 16-neighborhood search, although adding eight more search directions can improve the neighborhood continuity and make it more to meet the real operating state of the unmanned ship, which improves the search range and accuracy, more search directions also mean greater computational costs and require more computation time and storage space; in some cases, too wide a search range can also lead to some unnecessary information interfering with the calculation results and consuming computational resources. We adopt the 8-neighborhood search method in this paper.

### 2.3. Theory of Conventional A-Star Algorithm

A-star algorithm is a path planning algorithm proposed by Hart. Its basic idea is to use Heuristic search and Dij algorithm to find the shortest path from the initial point to the goal point. In the search process, the A-star algorithm maintains two values: the true cost g from the initial point to the current node, and the estimated cost h from the current node to the goal point. By calculating the total cost f of the current node, the priority of the node is evaluated, and the node with the smallest cost is preferentially extended until the goal point is found or the expansion cannot be continued. This can minimize the number of searches and can find the goal point as quickly as possible while ensuring the shortest path is found.

The evaluation function of the total cost is:
(1)$$f\left(n\right)=g\left(n\right)+h(n)$$
where, n represents the current node of the search; f(n) represents the total cost of the current node; g(n) represents the true cost from the initial point to the current node; h(n) represents the estimated cost from the current node to the goal point.

The A-star algorithm uses a heuristic function to estimate the cost h from the current node to the goal point. The commonly used heuristic functions, as shown in Figure 3, include the Manhattan distance, Euclidean distance, and Chebyshev distance.

The Manhattan distance formula is:(2)$$h\left(n\right)=\left|x_{n}-x_{G}\right|+\left|y_{n}-y_{G}\right|$$

The Euclidean distance formula is:(3)$$h\left(n\right)=\sqrt{{{(x}_{n}-x_{G})}^{2}+{{(y}_{n}-y_{G})}^{2}}$$

The Chebyshev distance formula is:(4)$$h\left(n\right)=\left(\sqrt{2}-2\right)\left(\min\left(\left|x_{n}-x_{G}\right|,\left|y_{n}-y_{G}\right|\right)\right)+\left|x_{n}-x_{G}\right|+\left|y_{n}-y_{G}\right|$$
where, $x_{n}$ and $y_{n}$ are the transverse and longitudinal coordinates of node n; $x_{G}$ and $y_{G}$ are the transverse and longitudinal coordinates of the goal point, respectively.

The specific implementation steps of the A-star algorithm are as follows:Initialization: initialize the initial point, the goal point, an open list, and a closed list. The Open list and the Closed list denote nodes that have been considered but not yet searched and nodes that have been searched, respectively. The initial point is placed in the Open list, its evaluation function f(n) is set to 0, and the parent node is set to null. Moreover, set the Closed list to empty.Iterative search: the node n with the smallest f(n) is taken out from the Open list each time and put into the Closed list. if n is the goal point, the algorithm terminates. Otherwise, all neighboring nodes of n are processed as follows:(1)If the neighboring node is already in the Closed list, ignore and skip.(2)If the neighboring node is not in the Open list, add it to the Open list and set its parent node to n. Moreover, calculate the evaluation function value of this neighboring node.(3)If the neighboring node is already in the Open list, compare the current evaluation function value of the neighboring node with the newly calculated evaluation function value. If the new value is smaller, update the parent node of the neighboring node to n and update its evaluation function value.Judge and terminate the search: when each node is traversed, judge whether the current node is the goal point, if so, indicate that the shortest path has been found, and terminate the search; if the Open list is empty, the goal point cannot be reached, the search fails, and terminate the search.Backtracking path: start from the goal point and follow the information of the parent node back to the initial point to get the shortest path.

## 3. Improved A-Star Algorithm

### 3.1. Improving the Evaluation Function

The evaluation function of the A-star algorithm consists of two parts, as shown in Formula (1), where g(n) can be usually assumed that the true cost of moving a grid laterally is 10, and the true cost of moving a grid diagonally is 14, and the specific value can be determined according to different circumstances; this paper adopts the 8-neighborhood search method, so the Euclidean distance formula, as shown in Formula (3), is used as the heuristic function to calculate the estimated cost h(n).

Based on the evaluation function of the A-star algorithm, the vertical distance between the node and the line connecting the initial point and the goal point is introduced as a deviation factor to improve the evaluation function. As shown in Figure 4, the red dot is the initial point S(x_S_, y_S_), the green dot is the goal point G (x_G_, y_G_), the yellow dot is the current node N(x_N_, y_N_), and the orange dot is the two neighboring nodes n_1_(x_1_, y_1_) and n_2_ (x_2_, y_2_) of the current node. The line between the initial point and the goal point is SG, and the vertical distance from the neighboring node n_i_ to SG is d(n_i_).

The linear equation of SG is:(5)Ax+By+C=0
where,
(6)$$\left\{\begin{array}{l} A=y_{G}-y_{S}\\ B=x_{S}-x_{G}\\ C=x_{G}y_{S}-y_{G}x_{S} \end{array}\right.$$

The calculation formula of vertical distance d(n_i_) is:(7)$$d\left(n_{i}\right)=\frac{\left|Ax_{i}+By_{i}+C\right|}{\sqrt{A^{2}+B^{2}}}$$

Therefore, the evaluation function of the current node n is:(8)$$f\left(n\right)=g\left(n\right)+\mu h^{\prime }(n)$$
(9)$$h^{\prime }\left(n\right)=\sqrt{{{(x}_{n}-x_{G})}^{2}+{{(y}_{n}-y_{G})}^{2}}+d\left(n\right)$$
where: $f\left(n\right)$, $g\left(n\right)$, $x_{n}$, $y_{n}$, $x_{G}$, and $y_{G}$ are defined the same as the formula in Section 2.3; $h^{\prime }(n)$ represents the estimated cost from the current node to the goal point of the improved method; $d\left(n\right)$ represents the vertical distance from the current node n to the line SG; μ represents the weight factor.

The new evaluation function adds d(n), which aims to make the A-star algorithm focus on searching the nodes near the line SG as much as possible when traversing the nodes. d(n_i_) is smaller, indicating that the neighboring nodes of the current node are closer to the line SG, and the neighboring nodes are given priority when traversing the nodes, so that the number of traversing nodes can be reduced and the computational speed can be improved, while the planning path converges to the line between the initial point and the goal point as much as possible.

### 3.2. Introducing the Bidirectional Search Strategy

Common search optimization strategies include pruning, iterative deepening, and bidirectional search. Among them, bidirectional search is applied to problems with well-defined initial and final states, and the search tree generated by both the search from the initial state and the reverse search from the final state can cover the entire state space. By using bi-directional search, the initial state and the final state are searched at the same time, resulting in two search trees that are halved in depth or breadth, and the answer can be obtained when they meet in the middle, which can effectively avoid the problem of traversing many nodes and computationally time-consuming, as shown in Figure 5.

The conventional A-star algorithm uses a one-way search, traversing nodes from the initial point until the goal point is found, which has many traversing nodes and is computationally time-consuming and inefficient. Therefore, a bidirectional search optimization strategy is introduced, in which the initial point and the goal point are used as target nodes for each other, and the search starts from the initial point and the goal point at the same time. The search from the initial point to the goal point is called forward search, and the search from the goal point to the initial point is called reverse search, whose search mechanism is shown in Figure 6.

In the path search process, the intersection of the nodes traversed by the forward search and the reverse search is set to:(10)J=Open list_S∩Open list_G
where: Open list_S represents the open list of forward search; Open list_G represents the open list of reverse search. When J is not the empty set, it means that the traversal nodes of forward search and reverse search intersect, that is, M in Figure 6. At this time, the search is terminated, and the planning path is obtained by starting from point M and going back to the initial point and goal point respectively according to the information of the parent node. When J is the empty set, it means the forward search and reverse search are unable to meet and the path planning fails. To solve the problem that bidirectional search cannot meet, Detong Chen [15] et al. introduced a dynamic window to establish a search dynamic window based on the distance difference between the transverse and longitudinal coordinates of two search nodes in the forward and reverse directions, and stop the search when the coordinate difference is less than a certain value at the same time; Sixin Wang et al. [16] used the gravitational idea of artificial potential field method to set the virtual goal point at the center of the bidirectional search and establish the distance decay function from the initial point of the search to the midpoint of the bidirectional search to attract the bidirectional search can meet in the middle region.

### 3.3. Smoothing the Planned Path

The planning path of the USV obtained by the improved A-star algorithm is a line graph, which does not match the actual navigation trajectory of the unmanned ship. Therefore, a spline curve is used—curves whose shape is controlled by a given series of restriction points, as shown in Figure 7. Common spline curves include: Bézier curves and B spline curves.

The B spline curve has all the advantages of the Bézier curve, but overcomes the disadvantages of the Bézier curve in that it cannot be modified locally and the difficulty of continuity condition, and has the characteristics of local adjustment, convex wrapping, continuity, and good path smoothing effect [17,18,19].

B spline curve of order k is defined as:(11)$$P\left(u\right)=\sum_{i=0}^{n}P_{i}B_{i,k}(u)$$
where: P_i_ represents the sequence of unmanned ship path nodes (restriction points), P_i_ = {P_1_, P_2,_ ⋯, P_i_, ⋯, P_n−1_, P_n_}; B_i,k_(u) is the i-th k-order B spline basis function, which corresponds to P_i_; i represents the B spline ordinal number, I = 0, 1, ⋯, n; k represents the order of B spline, k≥1; u represents the continuous variation in the non-decreasing sequence of the node vector, and the first and last values are generally defined as 0 and 1.

The B spline basis function has deBoor-Cox recursion:(12)$$B_{i,k}(u)=\left\{\begin{array}{l} \left\{\begin{array}{l} 1,\;u_{i}\leq u<u_{i+1}\\ 0,\text{ other} \end{array},\right.\;k=1\\ \frac{u-u_{i}}{u_{i+k-1}-u_{i}}B_{i,k-1}\left(u\right)+\frac{u_{i+k}-u}{u_{i+k}-u_{i+1}}B_{i+1,k-1}\left(u\right),\;k\geq 2\\ \text{Define}\frac{0}{0}=0 \end{array}\right.$$
where: u_i_ = [u_0_,u_1_, ⋯, u_k_, u_k+1_, ⋯, u_n_, u_n+1_, ⋯, u_n+k_].

Considering the time-consuming and costly computation, quadratic or cubic B spline curves are usually used; in addition, uniform B spline curves do not have geometric invariance at the endpoints of Bézier curves—the property that certain geometric characteristics do not change with coordinate transformation. In this paper, the cubic quasi-uniform B spline curve is used to smooth the planned path, as shown in Figure 8, which effectively solves the problem that the path is a folded line and has many inflection points, and reduces the path length to a certain extent.

### 3.4. Process and Pseudo-Code of the Improved A-Star Algorithm

Based on the implementation steps of the conventional A-star algorithm, a bidirectional search strategy is introduced, and a cubic quasi-uniform B spline curve is used to smooth the path after the algorithm obtains the planned path. The specific process is shown in Figure 9 and the pseudo-code is shown in the Appendix A.

## 4. Simulation Experiment and Analysis

To verify the feasibility and optimization effect of the textual algorithm, it is compiled on the PyCharm 2020.1 (Professional Edition) platform, simulated under different sizes of grid maps, and compared with the conventional A-star algorithm. The computer configuration used in the simulation experiment is: Windows 10 operating system, i7-1165G7 processor, 2.80 GHz main frequency, and 16 GB running memory.

This paper constructs grid maps of different sizes containing randomly set obstacles. In the figure, black dots are map boundaries and obstacles, white blanks are passable areas, purple dots are initial points, green dots are goal points, grass-green dots are forward searched traversed nodes, sky-blue dots are reverse searched traversed nodes, blue broken lines are the paths planned by the algorithm, and red curves are the final paths obtained by smoothing.

In the simulation experiment, the unmanned ship is usually regarded as a particle, that is, robot_radius = 0, which will cause the planned path to be too close to the obstacle, As shown in the red box in Figure 10a, resulting in the safety problem. Therefore, robot_radius = 1 is set in this paper, which is more to meet the safety needs of actual navigation, as shown in the red box in Figure 10b.

Figure 11a–c shows the simulation results of the conventional A-star algorithm, improving evaluation function and the use of bidirectional search; the specific data comparison is given in Table 1. It can be seen that: the improved evaluation function improves significantly in terms of the number of path inflection points, the maximum vertical distance, and the path length; while the bidirectional search improves significantly in terms of the total number of traversed nodes and the algorithm computation time. In the following section, the improved algorithm will be experimentally verified in different sizes of grid maps by combining the two improved methods.

Figure 12a,b shows the simulation results in a 40 × 40 grid map. It can be seen from the figure that the textual algorithm has fewer traversal nodes than the conventional one, and the efficiency of search is improved accordingly. In addition, the path is more convergent, but the number of inflection points has increased remarkably, so the path must be smoothed. The red curve in Figure 12c is the smoothed path, which is more to meet the actual navigation of the unmanned ship.

Since the inflection point of the planned path increased when the grid map was small, the grid map size was increased and simulation experiments were conducted in 60 × 60 and 80×80 raster maps, and the results are shown in Figure 13a,b and Figure 14a,b. From the figure, it can be seen that: at larger size maps, the traversal nodes are remarkably reduced, and the search time is correspondingly remarkably shortened, while the path inflection points are also reduced, but the paths also need to be smoothed, as shown in the red curves in Figure 13c and Figure 14c.

Table 2 and Table 3 give the specific data comparison of the above three sizes of grid map simulation experiments: when the map size is larger, the traversal node reduction ratio, the inflection point reduction ratio and algorithmic time reduction ratio are higher. It shows that the efficiency of search of the textual algorithm is improved, and the larger the map is, the more significant the improvement is. In addition, after the path smoothing process, the path length improvement rate is then significantly increased on top of the original improvement rate.

From the data in Table 2, it can be seen that in the 80 × 80 grid map, the path convergence is not remarkably improved. Therefore, we consider replacing the initial point and goal point and conducting simulation experiments, the results are shown in Figure 15, and the specific data comparison is shown in Table 2.

From the data in Table 4 and Table 5, it can be seen that after replacing the initial point and the goal point, the improved A-star algorithm still improves remarkably in the efficiency of search and the path length, the planning path also converges more to the line between the initial point and the goal point, which verifies the feasibility of the textual algorithm.

## 5. Conclusions

The conventional A-star algorithm has many problems in path planning, thus, this paper adopts a bidirectional search strategy to search nodes from the initial point and goal point in the meantime, and introduces a vertical distance node deviation factor based on the line connecting the initial point and goal point, and finally uses the cubic quasi-uniform B spline curve to smooth the planned path, remove the inflection points and solve the path folding problem, and proposes a global path planning algorithm for USV based on the improved A-star algorithm. The simulation experiments show that textual algorithm can generate a path with fewer inflection points, the path tends to be connected to the initial point and the goal point, the traversal nodes are remarkably reduced, and the efficiency is remarkably improved. The smoothed path is also more to meet the actual navigation demands of USV, which provides a better path planning method for USV to carry out environmental monitoring in inland waterways, reservoirs, or coastal scenes.

Although this paper proposes a new planning algorithm, it has the following deficiencies:This paper aims to provide a new path planning algorithm for the surface unmanned ship, but in the improvement process, mainly based on the theory of the conventional A-star algorithm method to optimize appropriately, for the unmanned ship kinematics parameters and other considerations are not well thought out, although the simulation experiment proves the algorithm’s superiority, but in the actual use of the unmanned ship characteristics need to be taken into account to make the appropriate adjustments.This paper verifies the algorithm as a simulation experiment, which theoretically verifies the feasibility of the algorithm. However, in the actual use of the unmanned ship, it will be affected by the environment and other comprehensive factors, so the reliability of the unmanned ship is required to be higher. Simulation experiments alone may not be able to meet the requirements of actual use, therefore, the subsequent need to carry out relevant real ship experiments to further verify the reliability of the algorithm.

## Figures and Tables

**Figure 1 sensors-23-06647-f001:**
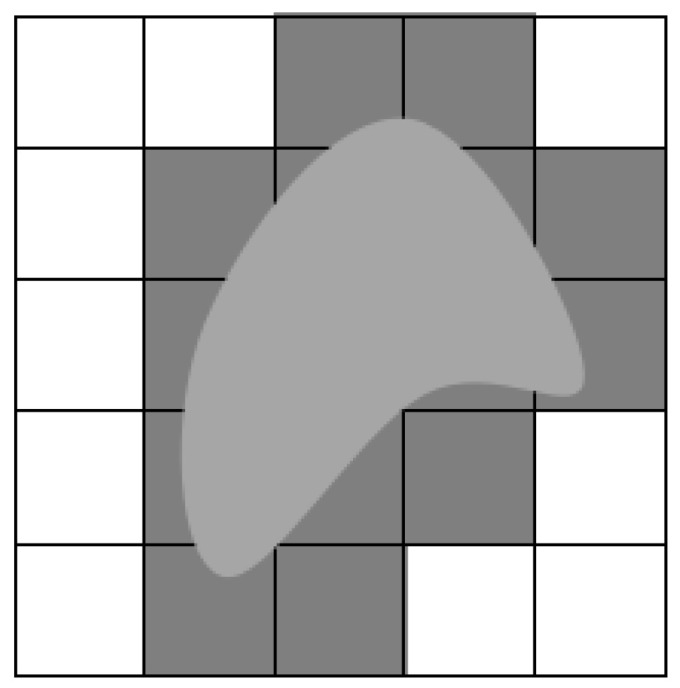
Grid map.

**Figure 2 sensors-23-06647-f002:**
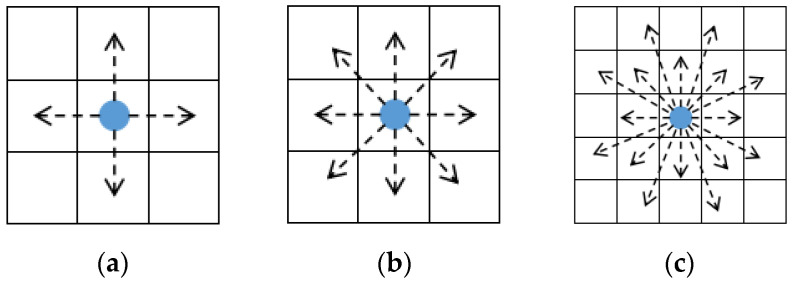
Diagram of different search neighborhood. The searches are: (**a**) 4-neighborhood search, (**b**)-8 neighborhood search, (**c**) 16-neighborhood search.

**Figure 3 sensors-23-06647-f003:**
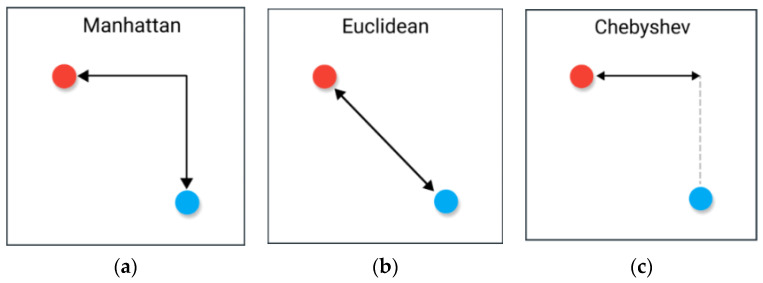
Diagram of a heuristic function. (**a**) Manhattan distance, (**b**) Euclidean distance, (**c**) Chebyshev distance.

**Figure 4 sensors-23-06647-f004:**
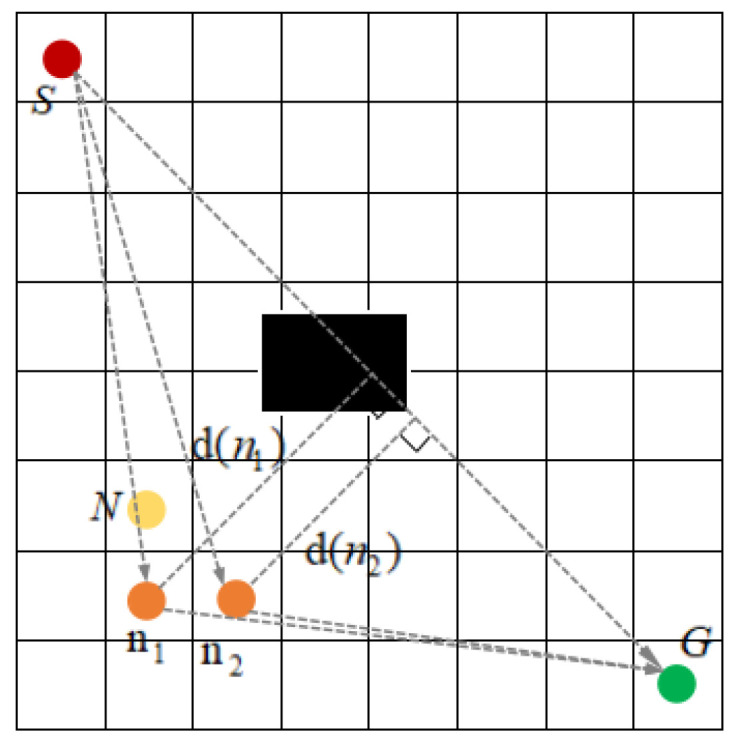
Diagram of node vertical distance.

**Figure 5 sensors-23-06647-f005:**
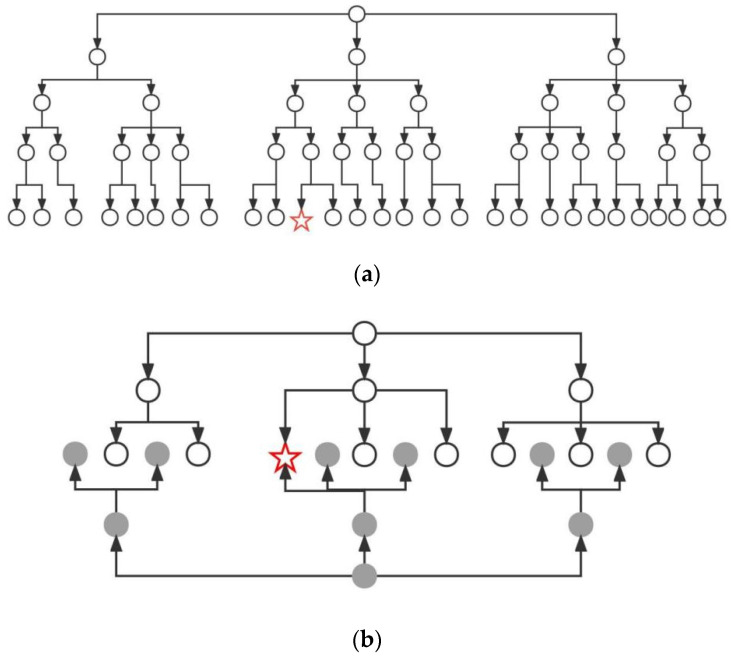
Search tree of different search methods. (**a**) Search tree for direct search, (**b**) search tree for bidirectional search.

**Figure 6 sensors-23-06647-f006:**
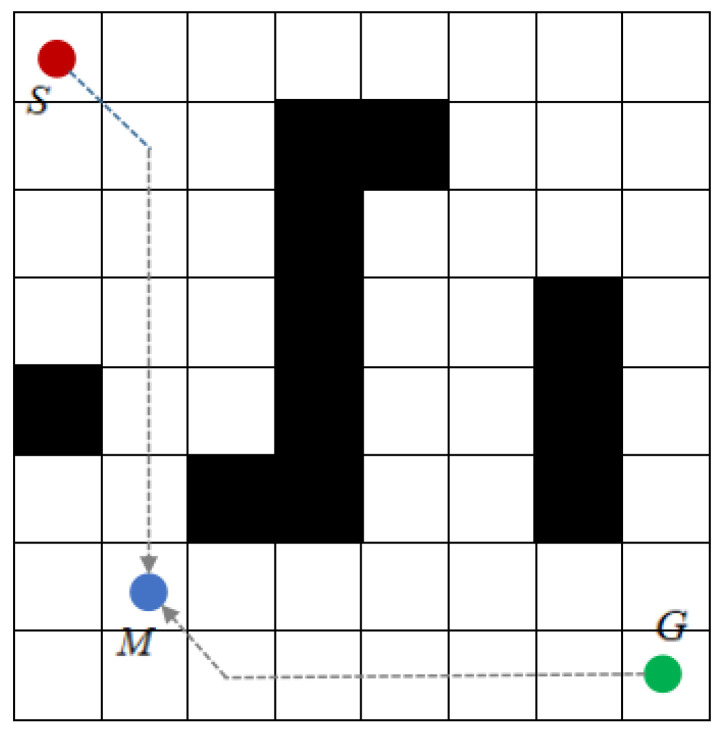
A-star algorithm bidirectional search mechanism.

**Figure 7 sensors-23-06647-f007:**
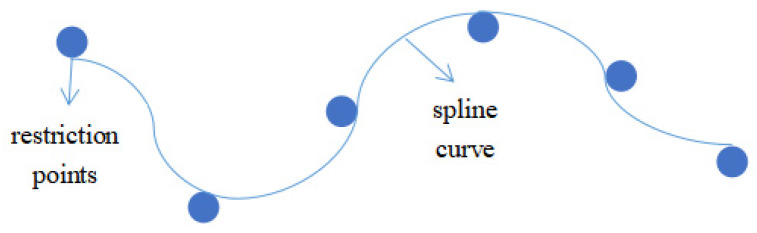
Diagram of spline curves.

**Figure 8 sensors-23-06647-f008:**
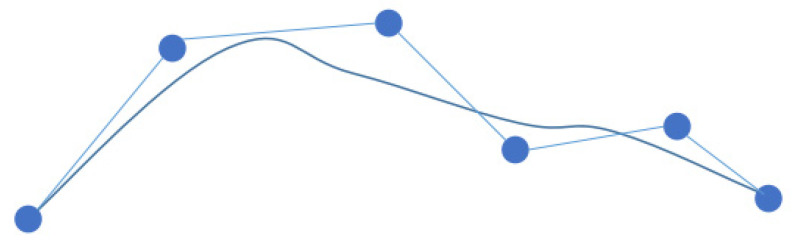
Diagram of cubic quasi-uniform B spline curve.

**Figure 9 sensors-23-06647-f009:**
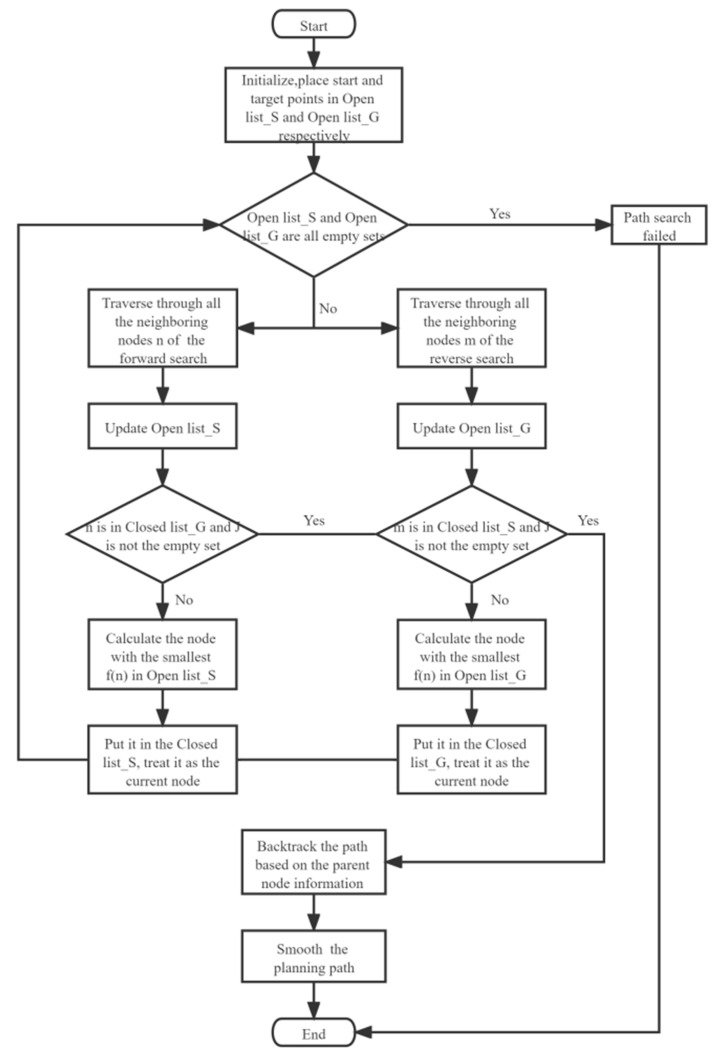
Path planning process of improved A-star algorithm.

**Figure 10 sensors-23-06647-f010:**
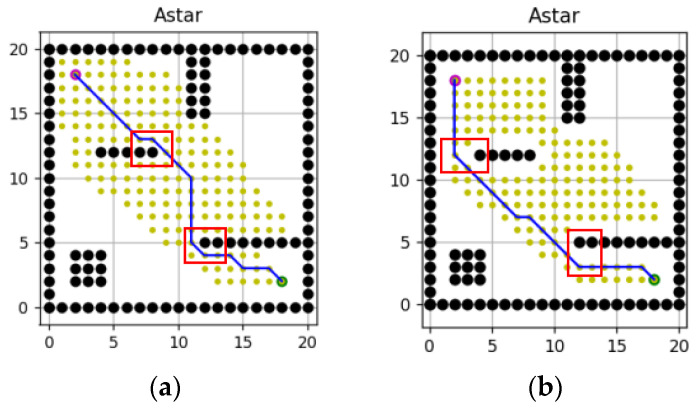
Diagram of path planning. (**a**) robot_radius = 0, (**b**) robot_radius = 1.

**Figure 11 sensors-23-06647-f011:**
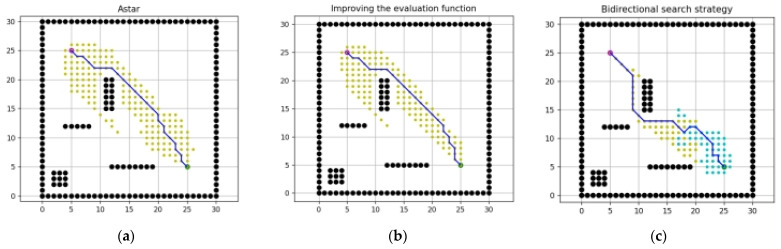
Simulation results of different methods. (**a**) The conventional A-star algorithm, (**b**) improving evaluation function, (**c**) bidirectional search.

**Figure 12 sensors-23-06647-f012:**
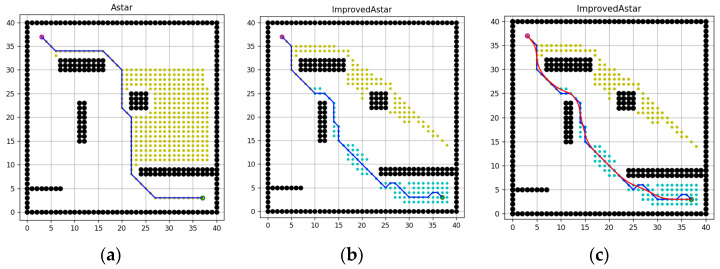
Simulation results of 40 × 40 map. (**a**) Path of the conventional A-star algorithm, (**b**) Path of the improved A-star algorithm, (**c**) Smoothed path.

**Figure 13 sensors-23-06647-f013:**
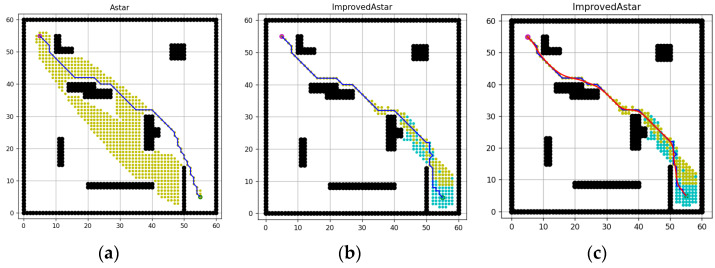
Simulation results of 60 × 60 map. (**a**) Path of the conventional A-star algorithm, (**b**) Path of the improved A-star algorithm, (**c**) Smoothed path.

**Figure 14 sensors-23-06647-f014:**
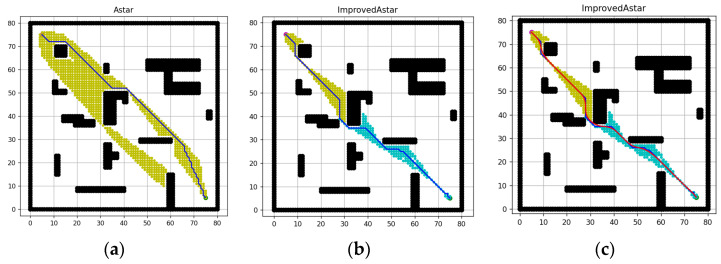
Simulation results of 80 × 80 map. (**a**) Path of the conventional A-star algorithm, (**b**) Path of the improved A-star algorithm, (**c**) Smoothed path.

**Figure 15 sensors-23-06647-f015:**
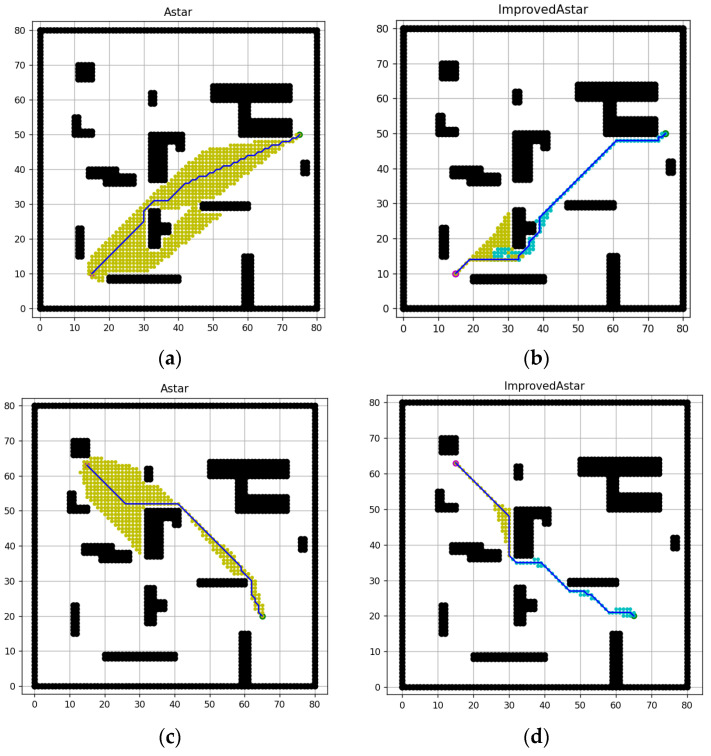
Simulation results of different initial points and goal points of 80 × 80 map. (**a**) Path of the conventional A-star algorithm ((15, 10), (75, 50)), (**b**) Path of the improved A-star algorithm ((15, 10), (75, 50)), (**c**) Path of the conventional A-star algorithm ((15, 63), (65, 20)), (**d**) Path of the improved A-star algorithm ((15, 63), (65, 20)).

**Table 1 sensors-23-06647-t001:** Comparison of simulation results for different methods.

Method	Map Size/m	Coordinates of Starting Point and Target Point	Total Number of Nodes Traversed	Improving Ratio	Number of Path Inflection Points	Improving Ratio	Maximum Vertical Distance	Improving Ratio	Algorithm Computation Time/s	Improving Ratio	Length of Path/m	Improving Ratio
A* algorithm	30 × 30	(5, 25)	174		12		2.83		1.878		33.4	
		(25, 5)										
Improving evaluation function	30 × 30	(5, 25)	155	10.9%	10	16.7%	2.12	25.1%	1.806	3.8%	29.7	11.1%
		(25, 5)										
Bidirectional search	30 × 30	(5, 25)	102	41.4%	11	8.3%	2.83	0%	1.533	18.4%	34.7	−3.9%
		(25, 5)										

**Table 2 sensors-23-06647-t002:** Comparison of simulation results for different algorithms.

Map Name	Map Size/m	Coordinates of Starting Point and Target Point	Total Number of Nodes Traversed		Improving Ratio	Number of Path Inflection Points		Improving Ratio	Maximum Vertical Distance		Improving Ratio
			A* Algorithm	Improved A* Algorithm		A* Algorithm	Improved A* Algorithm		A* Algorithm	Improved A* Algorithm	
Map1	40 × 40	(3, 37)	342	210	37.6%	7	15	−114.3%	7.071	6.364	10.0%
		(37, 3)									
Map2	60 × 60	(5, 55)	611	225	63.2%	24	20	16.7%	19.799	18.385	7.1%
		(55, 5)									
Map3	80 × 80	(5, 75)	759	241	68.2%	10	7	30.0%	4.243	4.243	0.0%
		(75, 5)									

**Table 3 sensors-23-06647-t003:** Comparison of simulation results for different algorithms.

Map Name	Map Size/m	Coordinates of Starting Point and Target Point	Algorithm Computation Time/s		Improving Ratio	Length of Path/m			Improving Ratio	
			A* Algorithm	Improved A* Algorithm		A* Algorithm	Improved A* Algorithm	Path Smoothing		
Map1	40 × 40	(3, 37)	2.723	2.017	25.9%	59.7	57.5	51.1	3.7%	14.4%
		(37, 3)								
Map2	60 × 60	(5, 55)	4.683	2.673	42.9%	83.4	80.5	74.2	3.5%	11.0%
		(55, 5)								
Map3	80 × 80	(5, 75)	7.953	3.878	52.4%	112.5	109.1	100.5	3.0%	10.7%
		(75, 5)								

**Table 4 sensors-23-06647-t004:** Comparison of simulation results for different initial points and goal points.

Map Name	Map Size/m	Coordinates of Starting Point and Target Point	Total Number of Nodes Traversed		Improving Ratio	Number of Path Inflection Points		Improving Ratio	Maximum Vertical Distance		Improving Ratio
			A* Algorithm	Improved A* Algorithm		A* Algorithm	Improved A* Algorithm		A* Algorithm	Improved A* Algorithm	
Map3	80 × 80	(5, 75)	759	241	68.2%	10	7	30.0%	4.243	4.243	0.0%
		(75, 5)									
Map3	80 × 80	(15, 10)	728	179	75.4%	32	11	65.6%	7.49	6.66	11.1%
		(75, 50)									
Map3	80 × 80	(15, 63)	342	103	69.9%	10	9	10.0%	10.19	8.613	15.5%
		(65, 20)									

**Table 5 sensors-23-06647-t005:** Comparison of simulation results for different initial points and goal points.

Map Name	Map Size/m	Coordinates of Starting Point and Target Point	Algorithm Computation Time/s		Improving Ratio	Length of Path/m		Improving Ratio
			A* Algorithm	Improved A* Algorithm		A* Algorithm	Improved A* Algorithm	
Map3	80 × 80	(5, 75)	7.953	3.878	52.4%	112.5	109.1	3.0%
		(75, 5)						
Map3	80 × 80	(15, 10)	5.987	3.010	49.7%	80.0	77.5	3.1%
		(75, 50)						
Map3	80 × 80	(15, 63)	4.246	2.226	47.6%	72.5	70.1	3.3%
		(65, 20)						

## Data Availability

No new data were created or analyzed in this study. Data sharing is not applicable to this article.

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
