# Peer review of "Global Path Planning of Unmanned Surface Vehicle Based on Improved A-Star Algorithm"

_sensors, 2023, doi:10.3390/s23146647_

Round 1

Reviewer 1 Report

In the paper under review, the authors propose a path-planning algorithm for an unmanned surface vehicle in view of environmental monitoring. They use an improved A-star algorithm. They apply their approach by simulation.

My overall rate of the paper is as follows.

- Though the motivations are interesting and they well motivate the paper, the authors should explain much better how to include the environment model in the problem formulation.

- In general, the approach is too much qualitative and poorly quantitative. I mean, the authors explain too much through words and much less using formulas/mathematics. It really confuses me, I often do not understand what the authors exactly mean. The authors should be more clear and quantitative, they should use more mathematics to formulate and solve the problem. Consider that Figure 9 has the formalism of an algorithm, but an algorithm is fully described by formulas and not sentences. Please modify accordingly.

- Section 1.3 describes the conventional A-star algorithm. It is a well-known approach, but  Section 1.3 lacks of clarity and rigor. I think that it is useless in the present form and it can be omitted. Otherwise, my advice is to explain it clearly and with rigor. 

- The English is pretty good but there are a lot of sentences that are not much informative or questionable. As for example, the very beginning is: "Unmanned Surface Vehicle (USV) is a type of unmanned and autonomous navigation intelligent ship." which is not so clear to me. Or: ..."which effectively solves the problem that the paths is folded and has many inflection points...".

- A practical example of the specific application (a practical marine application) would improve the overall paper quality.

English is pretty good from a grammar perspective, but there are some errors here and there (e.g. in the abstract: "we proposed a global path planning algorithm based 7 on the improved A-star algorithm" why the past? Did you propose in the past or are you proposing in this paper???). 

Author Response

Dear Reviewer,

Thank you for the review report. For your comments and suggestions, I have made changes in the paper or responded in the attached Respond Reviewer. 

Sincerely yours,

The author

Reviewer 2 Report

1. The cost function of the A star is composed of g (n) and h (n). In combination, they already include the idea of the shortest total path, which will achieve the purpose of preferentially searching for the nearby area of the SG line segment. So you separately strengthen the meaning of this layer in the evaluation function h (n), is it more meaningful?

2. I don't care too much about the improvement of the A-star algorithm when you combine the two-way search method and the spline method. Because the fusion of bidirectional search is already available, the spline is just the icing on the cake. So your experiment should focus on comparing the improvement of your evaluation function to the A star algorithm, not only with the evaluation function of the classic A-star algorithm, but also with the excellent evaluation function of others.

Author Response

Dear Reviewer,

Thank you for the review report. For your comments and suggestions, I have made changes in the paper or responded in the attached Respond Reviewer. 

Sincerely yours,

The authors

Reviewer 3 Report

This paper proposes a global path-planning method for USV. Some modifications should be made before publication:

1) For vehicle path planning, obtaining the surrounding obstacle information is essential. The following work provides the pipeline for the surrounding environment perception: an automated driving systems data acquisition and analytics platform, hydro-3d: hybrid object detection and tracking for cooperative perception using 3d lidar, yolov5-tassel: detecting tassels in rgb uav imagery with improved yolov5 based on transfer learning. It is meaningful to include and discuss the above work in the introduction from sensor configuration, including GNSS, LiDAR, and camera.

2) For the improved A* algorithm, the system’s location and speed are the input. Unfortunately, these signals are always polluted by noise and bias. Preprocessing the raw signal is important. The following work provides robust algorithms to solve this issue: autonomous vehicle kinematics and dynamics synthesis for sideslip angle estimation based on consensus kalman filter, vehicle sideslip angle estimation by fusing inertial measurement unit and global navigation satellite system with heading alignment, imu-based automated vehicle body sideslip angle and attitude estimation aided by gnss using parallel adaptive kalman filters, automated vehicle sideslip angle estimation considering signal measurement characteristic. Thus, you should add a part to introduce the Kalman filter and its variants including the above work to support your precise signal input.

3) The box text in Figure 9 expresses too much and needs to be refined.

4) Please provide the algorithm computation time.

5) Please highlight the work contribution in the introduction and list the work limitation and future work at the end.

6) As you have introduced two major modifications, the ablation study should be provided for each modification to verify the effectiveness of the corresponding modifications.

Author Response

(The authors gave the same response as above.)

Reviewer 4 Report

It is not clear to clarify and understand that which kind of ship mobile robot has been employed to analyse the path planning on the sea or river with improved A-star algorithm.

 Kinematic and constant parameters should be given with schematic view of the unmanned ship system. Also, it should be described in the text of the paper. 

As can be seen in the Figure 9, improved A-star algorithm should be outlined in the text with broad description for good understanding the improved strategy. In the text algorithm was described very short without good description.  

The unmanned ship phrase emphasized throughout the article is not descriptive enough. Can't the study be done for every planar autonomous vehicle as well?

While smoothing the planned path, why is the method used in curve fitting operations not compared with other methods? The advantages and disadvantages that affect the method selection are detailed but not presented with supporting references.

The numerical values of the parameter(s) used in calculating the B-Spline curve are not expressed.

If the pseudo-code of the path planning process with the improved A-star algorithm was given and the steps were explained, it would be both a more up-to-date presentation style and more understandable.

It should be explained what the red boxes in Figure 10 represent. The expression red curves before Figure 10 can be confused with these boxes. In addition, these expressions only seem to point to Figure 10.

How the radius change, explained in Figure 10, is carried out in the algorithm. So, how do we stay clear of obstacles in Figure 10.b?

Computation time is significant in path-planning algorithms. It is important to share this information to reveal the study's general validity and originality. However, the calculation times of the traditional and proposed methods are not given and compared in the study. It would also be helpful to present the path lengths.

Curve fitting processes were not performed for the traditional method. Therefore, the smoothed paths for the proposed method have not been compared with the traditional method with any success criteria.

Since smoothing the planned path, what is the advantage of reducing the number of path inflection points?

It has not been examined why the proposed method gives better results regarding "number of path inflection points" as the map size increases.

While the improvements have high percentages regarding "number of path inflection points", isn't the percentage of change quite small considering the total number of points?

Although the maps in Figure 14 are 80x80, a 60x60 part of the map may be used considering the paths covered. From this point of view, how accurate is it to compare this figure with Figure 13?

Paper should be improved with good discussion section.

Technical sound of the paper is weak . it should be improved with good English writing

Author Response

(The authors gave the same response as above.)

Round 2

Reviewer 2 Report

1. The flowchart in Figure 9 is not clear enough, and the direction of some processes is not indicated.

2.  The image in Figure 11 is too blurry, please redraw it.

3. The experimental data are too little and accidental. It is suggested that the start and end points should be randomly selected several times on each map, and then the data should be collated.

Author Response

Dear Reviewer,

Thank you for the review report again. For your comments and suggestions, I have made changes in the paper or responded in the attached Respond Reviewer. 

Sincerely yours,

The author

Reviewer 4 Report

Fig.11(a) (b) (c)shows the simulation results of the conventional A-star algorithm, improving evaluation function  and the use of bidirectional search ; the specific data comparison is given in Tab. 1. It can be seen that: the improved  evaluation function improves significantly in terms of the number of path inflection points, the maximum vertical  distance and the path length; while the bidirectional search improves significantly in terms of the total number of raversed nodes and the algorithm computation time. In the following section, the improved algorithm will be experimentally verified in different sizes of grid maps by combining the two improved methods. 

This description should be under Figure 11.

The technical sound of the paper should be improved with good English.

Author Response

(The authors gave the same response as above.)
